# May I Help You with Your Coat? HIV-1 Capsid Uncoating and Reverse Transcription

**DOI:** 10.3390/ijms25137167

**Published:** 2024-06-28

**Authors:** Laura Arribas, Luis Menéndez-Arias, Gilberto Betancor

**Affiliations:** 1Instituto Universitario de Investigaciones Biomédicas y Sanitarias (IUIBS), Universidad de Las Palmas de Gran Canaria, 35016 Las Palmas de Gran Canaria, Spain; laura.arribas@ulpgc.es; 2Centro de Biología Molecular “Severo Ochoa” (Consejo Superior de Investigaciones Científicas & Universidad Autónoma de Madrid), 28049 Madrid, Spain; lmenendez@cbm.csic.es

**Keywords:** HIV-1, infection, capsid, uncoating, reverse transcription, nuclear pore

## Abstract

The human immunodeficiency virus type 1 (HIV-1) capsid is a protein core formed by multiple copies of the viral capsid (CA) protein. Inside the capsid, HIV-1 harbours all the viral components required for replication, including the genomic RNA and viral enzymes reverse transcriptase (RT) and integrase (IN). Upon infection, the RT transforms the genomic RNA into a double-stranded DNA molecule that is subsequently integrated into the host chromosome by IN. For this to happen, the viral capsid must open and release the viral DNA, in a process known as uncoating. Capsid plays a key role during the initial stages of HIV-1 replication; therefore, its stability is intimately related to infection efficiency, and untimely uncoating results in reverse transcription defects. How and where uncoating takes place and its relationship with reverse transcription is not fully understood, but the recent development of novel biochemical and cellular approaches has provided unprecedented detail on these processes. In this review, we present the latest findings on the intricate link between capsid stability, reverse transcription and uncoating, the different models proposed over the years for capsid uncoating, and the role played by other cellular factors on these processes.

## 1. Introduction

The discovery of the dogma-breaking concept of reverse transcription by Howard Temin and David Baltimore in 1970 demonstrated that genetic information can flow from RNA to DNA. This process is exploited by different viral families, such as *Caulimoviridae* [1] or *Hepadnaviridae* [2], but is best known from retroviruses, which inherited their name from this process. The most studied member of this family is human immunodeficiency virus type 1 (HIV-1). HIV-1 is a major global health threat, infecting 39 million people worldwide [3].

Over 40 years of intensive research has uncovered many aspects of HIV-1 biology [4]. Therefore, we know that HIV-1 is an enveloped virus decorated with viral glycoproteins (Env), responsible for recognizing specific cellular receptors [5]. Surrounded by the viral envelope resides the capsid. This container-like structure, characterized by its roughly conical shape, is built by around 1600 copies of a single protein, also called capsid (CA), arranged in hexamers and pentamers [6,7,8,9,10,11]. This protein core contains all the viral components required for replication, including two copies of the viral genomic RNA and the enzymes reverse transcriptase (RT) and integrase (IN).

RT transforms the genomic RNA into a double-stranded DNA molecule [proviral DNA] that is transported into the cell nucleus where IN mediates its integration into the host chromosome. There, the proviral DNA (now called provirus) remains stably integrated, resulting in the transfer of the provirus from the parental to all daughter cells.

The relevance of capsid during HIV-1 infection was not initially appreciated and it was viewed simply as a vehicle for transporting the viral RNA and proteins [12]. Nevertheless, its crucial role in viral replication began to be recognized through early studies of its interactions with cellular components. Therefore, in the early 1990s, several groups identified a cellular protein, Cyclophilin A (CypA), which interacts with a motif in the viral capsid, named the CypA binding loop, on its honor, and that is incorporated into viral particles [13,14,15,16]. Concomitantly, several studies focused on the determination of capsid structure (reviewed in [17]), interaction with other cellular partners (reviewed in [18]), or maturation (reviewed in [19]). Nevertheless, a concept that was accepted early as inevitable was that capsid needs to disassemble, in a process termed uncoating, to release the viral components, access cytoplasmic dNTPs, and thus enable the progression of the viral life cycle.

On the other hand, RT was early recognized as a key player in viral replication and was therefore the target of intense research. This led to the characterization of the process of reverse transcription (discussed below), and the design of diverse compounds with the ability of inhibiting RT function. The wealth of HIV-1 inhibitors targeting the RT and other viral enzymes resulted in combination therapies capable of suppressing viral replication, referred to as highly active antiretroviral therapy (HAART). The inception of HAART marked an important turning point in HIV-1 infection treatment and led to a dramatic increase in the survival rate of HIV-1-infected individuals [20]; for updated reviews on the state of RT inhibitors, please refer to [21,22,23]. However, both processes (i.e., reverse transcription and capsid uncoating) were most frequently studied in isolation, mostly owing to technical limitations, which for years hindered the recognition of their intricate nature.

In this review, we will discuss the relationship between HIV-1 reverse transcription and capsid uncoating. Existing data on the timing, location, and how the progress of one affects the other will be presented. Finally, we will enumerate several unanswered questions in the field.

## 2. The Process of Reverse Transcription

The replication of the HIV-1 genome involves the conversion of single-stranded RNA (ssRNA) into double-stranded DNA (dsDNA) carried out by the viral RT. The RT is a heterodimeric enzyme composed of subunits p66 and p51, and of 66 kDa and 51 kDa, respectively. The p51 subunit is a truncated form of p66, lacking the RNase H domain formed by residues 441–560. Both subunits contain four subdomains termed “fingers” (amino acids 1–85 and 118–155), “palm” (86–117 and 156–236), “thumb” (237–318), and “connection” (319–426) [24]. However, these subdomains fold in different configurations in each subunit. The p51 subunit plays a structural role, while the DNA polymerase active site, formed by residues Asp110, Asp185, and Asp186, resides in p66.

While RT DNA polymerase and RNase H activities are sufficient to complete all the steps in the reverse transcription process, the viral nucleocapsid (NC) protein is a necessary partner; for a recent review, see [25]. The mature NC protein is a nucleic acid-binding protein with a chaperoning role in reverse transcription [26]. NC is involved in the initiation of reverse transcription by promoting hybridization of the cellular transfer RNA (tRNA) to the primer binding site (PBS) in the viral genome. In addition, it has a role in facilitating strand transfer events occurring during the process. In vitro, NC increases RT processivity and the fidelity of DNA synthesis [27]. However, the impact of those effects in vivo is still unknown. The role of NC in the viral replication cycle relates to the structural reorganization of the viral genomic RNA structure. NC acts as a chaperone, by modulating nucleic acid rearrangements and thereby facilitating the formation of structures which are thermodynamically more stable [28].

The HIV genome is made of two positive-sense ssRNA molecules with a structural organization that resembles cellular messenger RNA (mRNA). Thus, the viral genomic RNA contains a 5′ cap and a polyadenylated tail. The product of the reverse transcription process is a proviral dsDNA which is flanked by LTRs. HIV-1 LTRs have around 634 base pairs, segmented into three regions, known as U3 (unique 3′), R (repeat), and U5 (unique 5′). LTR ends participate in the integration of the dsDNA into the host genome. 

The reverse transcription process is outlined in Figure 1. Reverse transcription initiates after the binding of a cognate cellular tRNA to the PBS. The PBS is positioned downstream of the unique 5′ (U5) region, in the untranslated leader region. The 3′-OH of the tRNA serves as a primer for RNA-dependent DNA synthesis. All lentiviruses (including HIV) use tRNA^Lys3^ as a reverse transcription initiation primer. In HIV-1, the 3′-terminal 18 nucleotides of the tRNA primer anneal the complementary PBS (positions +182 to +199) found in the viral genome. Structures of RT initiation complexes have been recently obtained using cryo-electron microscopy [29] and by crystallography [30]. In the structure obtained by cryo-electron microscopy, authors introduced a covalent cross link between the enzyme and the full-length tRNA^Lys3^ annealed to 101 nucleotides of the viral 5′ UTR, while Das et al. [30] reported the structure of HIV-1 RT complexed with 23 nucleotides of the viral genomic RNA annealed to 17 nucleotides of the tRNA. 

Although both RTs are catalytically active, none of the structures captures the initiation complex poised for nucleotide incorporation. The 3′ primer terminus is displaced from the active site by 5–7 Å while the fingers subdomain of the RT remains open [32]. This is consistent with studies showing the slow polymerization rate shown by HIV-1 RT while incorporating the first 5–6 nucleotides at the 3′ end of the tRNA primer [33]. After these incorporations, the process speeds up considerably [34].

HIV virions contain two copies of genomic RNA and an estimated 20–770 copies of tRNA [35,36]. During reverse transcription, an intermediate known as minus-strand strong-stop DNA [(−)ssDNA] is formed once the tRNA primer is extended to the 5′ end of the viral genome. The 5′ end of the viral RNA is then removed by the RNase H activity of the RT, and the newly synthesised (−)ssDNA becomes available (Figure 1, second diagram). Duplicated sequences known as repeats (R) are found at both ends of the viral genomic RNA. The R sequence in the (−)ssDNA facilitates the strand transfer event by which DNA synthesis can continue from the 3′ of the viral RNA. The tRNA participates in both the first (−) and the second (+) strand transfer events [37].

Minus-strand DNA synthesis can resume on the same RNA or on a different RNA template and its elongation continues with simultaneous degradation of the template RNA (catalyzed by the RNase H activity of the RT). However, all retroviruses contain at least one purine-rich sequence at the 3′ region of the RNA genome (U3 region), but HIV-1 and other retroviruses have an additional polypurine tract (PPT) located in the central part of the genome (cPPT). PPTs are sequences of 15 ribonucleotides, that in HIV-1 contain a stretch of eight adenines with a single intervening guanine and a stretch of six guanines. The 5′ end of the PPTs is flanked by uridine-rich regions that, together with the intervening guanine, evolved to protect the integrity of the downstream PPT sequence [38,39].

PPTs are resilient to cleavage by the RT’s RNase H activity, but their formation relies on the introduction of specific cuts at its termini (within the U-tract and at the PPT-U3 junction). The specificity of these cleavages depends on the distance from the DNA polymerase and RNase H catalytic sites within the viral RT and can be influenced by mutations at the connection subdomain of the RT [40,41]. 

PPTs serve as primers for plus-strand DNA synthesis (Figure 1, plus strand DNA synthesis diagram). PPTs are extended to the 5′-end of the minus-strand DNA. When the growing DNA chain reaches the 18th nucleotide in the tRNA, polymerization is blocked due to the presence of a methylated base [42]. PPT elongation leads to the formation of plus-strand strong-stop DNA [(+)ssDNA]. When the process is completed, the tRNA is removed by the RT’s RNase H activity. This cleavage occurs at the 3′ end of the tRNA. In HIV-1, the completed minus-strand DNA contains an extra adenosine monophosphate (riboA) at its 5′ end [43,44]. The annealing of the (+)ssDNA to the 3′ end of the full-length minus-strand DNA involves a strand transfer reaction that occurs via base pairing of the complementary PBS sequences [45]. Finally, the strand displacement activity of HIV RTs facilitates the completion of the process and the formation of a full-length, integration-competent, double-stranded DNA with two identical LTRs, one at each end. Interestingly, mutations affecting the PPT were found in some patients treated with second-generation integrase inhibitors, such as dolutegravir. Biochemical studies showed that the mutated 3′-PPT sequence is unable to initiate plus-strand cDNA synthesis. Since this priming event is critical for the second strand transfer reaction, its abrogation may cause the formation and accumulation of 1-LTR circles, and not the linear DNA [46].

In HIV, the presence of a central PPT in the viral genome leads to the formation of flaps of unknown function (Figure 1, bottom diagram). These flaps are constituted by three-stranded structures with overlapping positive-strand sequences. The strand displacement activity of HIV RT is responsible for the generation of those flaps, which are removed by cellular endonucleases. After flap removal, the activity of cellular ligases joining the DNA ends is required to generate a complete provirus that can eventually integrate in the host cell genome. 

## 3. Early Understanding of Uncoating and Its Relationship with Reverse Transcription

Efforts to characterize the nature of HIV-1 replication intermediates resulted in the identification of two different complexes formed during the early stages of viral replication. The first one to be assembled is the reverse transcription complex (RTC), followed by its maturation into a preintegration complex (PIC). Characterization of these intermediates heavily relied on biochemical analysis of isolated viral DNA and associated proteins. These techniques usually employed high multiplicity of infection and the lysis of the infected cells using harsh lysis buffers, followed by isolation of RTCs and PICs by sucrose gradient sedimentation. Several studies determined that RTCs/PICs were characterized by the presence of viral RNA and DNA, IN, and other viral proteins such as Vpr and matrix (MA) [47,48,49,50,51]. These complexes contained low amounts of RT but were competent for reverse transcription and integration in vitro [52,53,54]. Intriguingly, CA was barely detected or not detected at all in any of these complexes, leading to the logical conclusion that CA was lost soon after the virus fused with the cell membrane.

Similarly, early studies using purified HIV-1 virions showed that intact particles or isolated cores (that is, virions stripped of their envelope) can produce early reverse transcription products upon addition of dNTPs, in reactions dubbed natural endogenous reverse transcription (NERT) and endogenous reverse transcription (ERT), respectively [55,56,57,58]. However, these reactions very rarely progressed to full-length viral DNA, and this only happened when detergents were added, pointing again to the need for capsid disassembly for reverse transcription to progress [59].

However, some observations argued against this idea. A direct link was established between CA functionality and reverse transcription through the identification of CA mutations affecting capsid stability, which resulted in deficient synthesis of viral DNA products [60,61,62]. In addition, the identification of rhesus macaque restriction factor tripartite motif-containing protein 5 alpha (TRIM5α) [63], which requires assembled capsid lattices for viral inhibition [64], demonstrated that complete or nearly intact cores remain in the cell cytoplasm for some time after entry, where they are recognized and degraded by this protein [65,66,67,68]. Interestingly, TRIM5α-directed inhibition resulted in the accumulation of early reverse transcription products and the concomitant absence of late products [63]. This discovery demonstrated the dependence of reverse transcription on capsid stability, and it was reminiscent of the defect on reverse transcription observed with hypostable capsids [62]. Further support for the persistence of assembled capsids after cell entry came from early fluorescent microscopy experiments following individual HIV-1 particles in infected cells. These studies detected large amounts of CA associated with these particles as they moved through the cell cytoplasm [69].

Another inconsistency with the idea of rapid and complete shedding of viral CA upon cell entry is the notion that viral nucleic acids must be protected from cytosolic DNA sensors such as cyclic GMP-AMP (cGAMP) synthase (cGAS) and IFI16 [70,71], therefore avoiding the cellular innate immune response.

Studies on the reciprocal effect, i.e., the role of reverse transcription on capsid disassembly, came later, with a seminal work by Arhel et al., where the authors proposed that the completion of reverse transcription, with the subsequent formation of a DNA flap, is the trigger for capsid uncoating [72]. This conclusion was supported by further studies demonstrating that the inhibition of reverse transcription by chemical or genetic means delays uncoating [73], increasing capsid stability [74].

Nevertheless, early studies demonstrated that capsid uncoating and reverse transcription are interconnected phenomena, with the modulation of one influencing the other.

## 4. Capsid and Its Journey through the Cell

HIV-1 capsid is composed of multiple copies of CA, arranged in around 250 hexamers and 12 pentamers. While the hexamers form the bulk of this structure, the pentamers are essential to confer the needed curvature to close both ends of the core. This specialized structure defines multiple binding sites for cellular factors and small molecules to interact with [Figure 2]: (i) the CypA binding loop spans residues 88–90 and it is essential for Cyp-like domain-containing proteins binding, such as CypA [75] or nucleoporin (NUP) 358 [76]; (ii) the phenylalanine–glycine (FG) binding pocket is delimited by the amino-terminal domain–carboxy-terminal domain interface of adjacent CAs and is important for the binding of factors such as CPSF6 [77] or NUP153 [78], and small molecules with capsid stabilizing activity, including PF74 and lenacapavir [15,79]; (iii) the tri-hexamer interface lays on the boundary where three CA hexamers come together and is bound by cellular factors MX2 and NUP153 [80,81]; (iv) the central pore (also known as the R18 pore due to the central role of Arg18 on its function) is located at the centre of each CA hexamer (and perhaps pentamer) and is bound by FEZ1 and IP6 [82,83,84]; (v) the macromolecular structure of the core itself, which is targeted by TRIM5α and TRIMCyp [66,85].

Mounting evidence of the presence and persistence of viral capsid during early HIV-1 replication spurred the development of biochemical and cellular assays to track viral particles as they moved through the infected cell cytoplasm. Different techniques applied on different cell lines resulted in four main models for capsid uncoating, and are summarized in Figure 3. As stated before, early biochemical approaches based on the isolation of RTCs detected little, if any, CA associated and therefore supported a model in which uncoating occurred early after virus internalization [73,85,91,92,93,94]. It is worth noting that a recent study supports some degree of cytoplasmic uncoating, based on the sensing of reverse transcription products by the cytoplasmic sensor cGAS [95]. Recent years have witnessed the emergence of advanced techniques to track the fate of individual viral particles. This includes microscopy-based techniques such as super-resolution microscopy, cryogenic electron microscopy (cryoEM), and correlative light and electron microscopy (CLEM), nucleic acids fluorescent labelling techniques such as EdU click-labelling or the ANCHOR system, and viral particle labelling techniques (discussed below). The almost general agreement amongst all the groups employing this vast array of techniques is that intact, or nearly intact capsids reach the nuclear envelope. Moreover, three different models have been proposed based on the location where uncoating takes place.

### 4.1. Uncoating at the Nuclear Pore Complex (NPC)

This model proposes that the viral capsid opens before the replication complex shuttles into the nucleus [95,96,97,98,99,100,101]. By labelling the virion-incorporated protein APOBEC3F and measuring its diffusion as a surrogate for capsid uncoating, the Pathak group reported a significant loss of CA subunits before nuclear translocation [96]. Fernández et al. proposed that HIV-1 capsid interacts with the nuclear envelope protein transportin 1, a β-karyopherin, and that this interaction promotes its uncoating [100]. However, the main supporter of this model has been the Melikyan group. In three different publications employing immunofluorescence microscopy approaches based on direct labelling of viral proteins and indirect labelling of viral capsids through its interaction with a tetrameric CypA-dsRed chimeric protein, they visualized the loss of capsid integrity (loss of CypA-dsRed signal) before nuclear import of replication complexes [98,99,101].

### 4.2. Uncoating in the Nucleus

Other groups propose that uncoating occurs after nuclear import [100,101,102,103,104,105,106,107,108,109,110,111]. This theory has been historically rejected because the nuclear pore has a diameter of around 40 nm [112,113], and therefore is not wide enough for the passage of HIV-1 capsids of roughly 60 nm diameter on the wide end [6,11]. However, this notion has been challenged by a recent study from the Krausslich lab, in which correlative light and electron microscopy combined with subtomogram averaging enabled visualization of HIV-1 capsids in infected cells with unprecedented detail [108]. Interestingly, by studying the morphology of nuclear pore complexes in intact cells, the authors measured nuclear pores wider than previously described, of around 64 nm, and therefore wide enough for intact capsids to traverse. Furthermore, they detected multiple capsids as they passed through nuclear pores and therefore concluded that intact capsids reach the nucleus of infected cells.

In support of this in vivo study, the Xiong group developed mimics of the nuclear pore consisting of a ring-shaped DNA framework decorated with cytoplasmic fragments from NUP358, nuclear pore channel motifs from NUP62, and nuclear lumen fragments from NUP153, and observed virus cores nailing through these structures and interacting with the NUP fragments in an orderly manner [110].

A recent study has proposed that not only do assembled capsids traverse the nuclear pore, but that they are essential for nuclear import [111]. In this study, the authors employed an in vitro-assembled cohesive FG phase mimicking the gel-like matrix formed by the FG-rich motifs of several NUPs in the nuclear pore channel and observed that the capsid lattice acts as a nuclear transport receptor, enabling its translocation into the nucleus. Similarly, a coarse-grained model of the human nuclear pore complex demonstrated that entire capsids translocate into the nucleus, and that this process requires certain capsid elasticity to accommodate the structural stress generated during translocation [112].

One of the most elegant approaches supporting nuclear uncoating came from the Campbell group, who fused the nuclear pore channel protein NUP62 to a drug-inducible dimerization domain to block the nuclear import of replication complexes. The authors determined that the capsid inhibitor PF74, which only binds to assembled capsid lattices, was able to inhibit infection hours after the virus was insensitive to nuclear pore blockade, indicating that assembled viral capsids reach the nucleus [102].

### 4.3. Capsid Remodelling during Nuclear Import (NPC Remodelling)

This model proposes that the viral capsid is structurally modified as it passes through the nuclear pore [114,115,116,117]. Blanco-Rodriguez and co-workers employed electron microscopy coupled with gold-labelling of capsids to detect CA complexes at both sides of the nuclear envelope, but with different staining properties, concluding that the capsid was remodelled during nuclear entry [115].

Another study used hyperstable CA mutants that were proficient for reverse transcription but became trapped at the nuclear pore. The authors linked the inability to complete nuclear import to the lack of flexibility of these capsids to be remodelled and therefore travel through the pore [117]. In experiments analysing the behaviour of viruses containing fused GFP-CA subunits, Francis and co-workers determined that remodelling of the capsid as it traverses the nuclear pore is necessary for productive infection [116]. 

In a recent study utilizing fluorescence fluctuation spectroscopy to examine the interaction between various FG-containing NUPs and in vitro assembled capsid-like structures, the authors postulated that the capsid’s structure might undergo alterations during passage through the nuclear pore complex. This hypothesis stems from the significant binding of multiple NUPs’ FG motifs to the capsid simultaneously [118].

Finally, other studies have detected CA-containing complexes inside the infected cell nucleus, but without details on their assembly state [119,120,121,122,123].

In addition to this direct method, indirect hints have suggested that whole capsids are mostly intact during their travel through the cytoplasm, and on docking at the nuclear envelope. As stated before, the discovery of TRIM5α demonstrated that assembled capsids must remain in the cytoplasm for inhibition to happen. Since this discovery, several other cytoplasmic proteins have been found to interact with assembled capsids. Therefore, recent studies identified secondary binding sites for the cytoplasmic protein CypA in the capsid lattice, expanding two hexameric assemblies [124,125]. Microtubule-associated proteins BICD2 and KIFB5 were identified as key cellular shuttle transporters responsible for conveying viral capsids through the microtubule network. Notably, their interaction requires intact capsids [126,127,128]. Similarly, the kinesin-1 adaptor protein FEZ1 binds the pore form at the centre of CA hexamers (R18 pore) to transport the capsid towards the cell nucleus [82,83]. Recently, Sec24C has been shown to co-traffic with capsid in the cytoplasm of infected cells, engaging with the FG-binding pocket [129].

Nuclear envelope proteins have also been identified as capsid interactors. Amongst them, NUP153 and MX2 require the presence of assembled lattices to engage on the tri-hexamer interface of capsid, demonstrating again the persistence of assembled cores up to nuclear pore docking. Furthermore, the nuclear protein CPSF6 plays an important role in determining the integration sites of proviral DNA [130], and its interaction with capsid also requires the presence of assembled cores.

In conclusion, the initial idea of HIV-1 capsid disassembling early after membrane fusion has been substituted by models that support its persistence to at least the engagement with the nuclear pore, with the possibility of arriving at the nuclear lumen intact or only partially open.

## 5. Cell Factors Modulating Reverse Transcription

Numerous cellular proteins play pivotal roles in regulating HIV-1 reverse transcription. Distinguishing between those directly influencing the reverse transcription process and those affecting capsid stability, and consequently viral DNA synthesis (like TRIM5α), can be challenging. Furthermore, in several cases, the factors listed here have been linked to both activities, reinforcing the notion of the intricate nature of reverse transcription and capsid uncoating.

### 5.1. APOBEC3 Proteins

Several apolipoprotein B editing complex members have shown antiviral activity [131]. These proteins are packaged into virions via interaction with the nucleocapsid protein and viral RNA [132,133,134,135]. Amongst them, APOBEC3G has been characterized as a potent inhibitor of retroviruses [136], retrotransposons [137], hepatitis B virus [138], and Vif-deficient HIV-1 [139]. APOBEC3G inhibits HIV-1 by a lethal mutagenesis mechanism in which deamination of deoxycytidine residues on the minus-strand DNA leads to inactivation of HIV-1 through guanine (G) to adenine (A) hypermutation of the proviral DNA [140,141,142,143]. Interestingly, in addition to inhibition by lethal mutagenesis, APOBEC3G has also been shown to restrict HIV-1 by blocking reverse transcription [144]. Although not fully understood, it has been proposed that the APOBEC3G-mediated inhibition of reverse transcription involves either interaction with template viral RNA or (−) ssDNA [145,146,147], or direct binding to the RT [148,149], in a mechanism involving cellular uracil base excision repair (UBER) enzymes [149].

Moreover, other protein members of the APOBEC3 family have also shown the ability to inhibit Vif-deficient HIV-1. APOBEC3F is a potent retroviral inhibitor producing G to A hypermutation in the proviral DNA [140,150]. In addition, APOBEC3F inhibits the accumulation of reverse transcription products in the absence of its hypermutation activity [150,151,152]. Finally, APOBEC3H seems to significantly inhibit reverse transcription in a mutagenesis-independent manner, accumulating late DNA products [153], in a mechanism involving the template switching frequency of the RT [154].

### 5.2. SAMHD1

Sterile alpha motif (SAM) and histidine-aspartic (HD) domain-containing protein 1 (SAMHD1) is a potent GTP/dNTP-stimulated triphosphohydrolase that converts deoxynucleoside triphosphates to the constituent deoxynucleoside and inorganic triphosphate [155,156]. In myeloid cells and resting T cells infected with HIV-1, SAMHD1 depletion leads to an increase in late reverse transcription products, indicating that SAMHD1 inhibits reverse transcription through depletion of dNTPs [157,158].

It is well established that phosphorylation of Thr592 negatively correlates with HIV-1 restriction by SAMHD1, and phosphomimetic mutants T592D and T592E abolish antiretroviral activity, but do not affect dNTP hydrolysis in vitro or in cellulo [159,160]. These findings raised the possibility of an HIV-1 inhibition mechanism independent of SAMHD1 dNTPase activity.

In this regard, it has been shown that the RNase activity of SAMHD1 degrades the viral RNA during early reverse transcription, inhibiting HIV-1 infection. Therefore, at low GTP concentrations, SAMHD1 exists as a monomer or a dimer with RNase but not dNTPase activity, while at higher GTP concentrations, the RNase activity of SAMHD1 is inhibited, promoting protein teramerization and its dNTPase activity [161]. Moreover, recent studies have focused on the nucleic acid binding activity of SAMHD1 and its role in HIV-1 inhibition. It has been shown that SAMHD1 can bind DNA and RNA oligomers in addition to GTP/dNTPs in the protein allosteric binding sites, resulting in active SAMHD1 tetrameric complexes with mixed occupancy of the binding sites [159,162].

### 5.3. p21

p21 [cyclin-dependent kinase inhibitor 1A], one of the p53 downstream-stimulated genes, is associated with cell cycle regulation, anti-apoptotic responses, and differentiation [163]. The expression of p21 inhibits the replication of HIV-1 and related primate lentiviruses in macrophages [164].

p21 restricts the expression of ribonucleotide reductase subunit R2 (RNR2), a key enzyme in dNTP biosynthesis, thus blocking dNTP production and consequently impairing reverse transcription [165,166,167]. In addition, it has been shown that downregulation of p21 strongly enhances the phosphorylation of SAMHD1, resulting in increased HIV-1 proviral DNA synthesis and virus infection, and strongly suggesting that p21 levels positively correlate with the extent of SAMHD1-mediated HIV-1 restriction [168]. Similarly, Shi et al. found that knockdown of p21 in human monocytes derived macrophages (hMDMs) increases SAMHD1 Thr592 phosphorylation, supporting that p21 regulates SAMHD1 phosphorylation in non-cycling cells [169].

Indeed, it has been shown that p21 is strongly upregulated in CD4^+^ T cells from elite controllers and reduces the susceptibility of these cells to HIV-1 by inhibiting viral reverse transcription [170,171].

### 5.4. Daxx 

Death-associated protein 6 (Daxx) is a highly conserved and ubiquitously expressed protein in mammals involved in numerous cellular processes, such as apoptosis, transcriptional repression, and carcinogenesis [172,173]. In addition, when recruited by promyelocytic leukemia protein (PML, also known as TRIM19), Daxx inhibits HIV-1 reverse transcription and retrotransposition of endogenous retroviruses [174].

It has recently been shown that a SUMO-interacting motif (SIM) located on the carboxi-terminal end of Daxx is required for its CypA-mediated interaction with capsid. Subsequently, Daxx would serve as a link for the recruitment of other cellular proteins such as TNPO3, TRIM5α, and TRIM34 to incoming HIV-1 cores. It has been proposed that Daxx–capsid interaction results in increased capsid stability, thus preventing uncoating and inhibiting reverse transcription [172].

### 5.5. eEF1A 

Human eukaryotic translation elongation factor 1A (eEF1A) is a protein subunit of the eukaryotic translation elongation 1 complex (eEF1) that has been described as an HIV-1 inhibitor [175].

eEF1A was identified as an RT cofactor from fractionated human T-cell lysates, whose depletion ablated the ability of these lysates to stimulate late reverse transcription in vitro. [176]. A recent study has uncovered that the RT thumb subdomain is involved in eEF1A binding, and that this interaction is key for HIV-1 uncoating, reverse transcription, and replication. Therefore, the mutation of surface-exposed acidic residues in the HIV-1 RT thumb domain, such as the conserved Glu300 residue, reduces RT interaction with eEF1A and leads to a delay in reverse transcription and uncoating kinetics [177].

### 5.6. Mov10

Moloney leukaemia virus 10 (Mov10) protein is an RNA helicase involved in the replication of endogenous retroviruses [178,179]. Like APOBEC3 proteins, Mov10 localizes to mRNA-processing bodies (P bodies) and is a component of the RNA-induced silencing complex (RISC) [180]. Several groups have shown that Mov10 overexpression inhibits HIV-1 infection at multiple stages of replication including virus production, proteolytic processing of Gag, and reverse transcription [180,181,182,183]. However, the mechanism behind these inhibitory activities is currently unknown.

### 5.7. RHA 

DHX9/RNA helicase A (RHA) is a DEAH-box helicase with RNA- and DNA-dependent helicase activity [184]. RHA is packaged into viral particles and enhances infectivity and reverse transcription [185,186,187,188]. Different mechanisms have been proposed for its ability to promote viral DNA synthesis, including increased tRNA^Lys3^ annealing [187], or (−)cDNA synthesis [185]. However, a recent study has found that RHA does not enhance DNA synthesis in the initial elongation steps, but rather increases RT processivity, resulting in more efficient reverse transcription [189].

### 5.8. TOP1

Topoisomerase 1 (TOP1) is a DNA topoisomerase that regulates the DNA topology during transcription [190]. Several studies have found that TOP1 enhances HIV-1 cDNA synthesis [191,192,193], and this has been related to the ability of TOP1 to dissociate the RT from structured RNA [194]. Specifically, TOP1 fostered in vitro reverse transcription reactions, a finding that was supported by the ablation of this effect upon addition of the TOP1 inhibitor camptothecin [191]. However, more recent studies have found that the antiviral activity of this drug is TOP1-independent [195].

## 6. The Timing, Location, and Effect of Reverse Transcription

Newly formed HIV-1 particles can incorporate dNTPs into their capsid. Therefore, reverse transcription could theoretically start in intact virions, and this was demonstrated by the identification of early reverse transcription products in viral particles [56,73]. Once inside the infected cell, reverse transcription continues until synthesis of full-length proviral DNA around 8–10 h after infection in immortalized cell lines [73,104,196,197], and around 12 h after infection in primary CD4^+^ T cells [198].

As pointed out before, it was widely accepted that viral DNA synthesis could only be completed upon capsid shedding, since the RTC would need access to more dNTPs present in the cell cytoplasm [199]. This notion changed in 2016 when the James lab showed that the pore existing in the centre of capsid hexamers (the R18 pore, Figure 2) allowed the import of dNTPs into the capsid [200]. Interestingly, it was later shown that this same pore is occupied by the cellular polyanion inositol hexakiphosphate (IP6), which promotes assembly of the mature capsid [84,201,202], increases stability [203,204,205], aids avoiding host immune sensing [206], facilitates dNTP import [207], and promotes ERT [196,197].

Proviral DNA constitutes a larger, more rigid molecule than RNA. The volume available inside the capsid is limited, and therefore, it is possible that synthesis of full-length reverse transcription products can only be completed after capsid opening. This notion seemed to be supported by the fact that ERT and NERT reactions can only proceed when detergents are added to the mixture, likely shedding the capsid lattice [59], and by more recent studies identifying the chaperone protein Daxx as an inhibitor of reverse transcription, via the inhibition of uncoating [172,174]. However, many studies have recorded active reverse transcription in closed cores. Therefore, ERT reactions carried out in the presence of IP6 are greatly enhanced in the production of late reverse transcription products, and hyperstable capsid mutants can complete viral DNA synthesis [62,117,196,203,208,209]. 

A hypothesis has been proposed to explain the advantages of reverse transcribing the genomic RNA into closed capsids, which is commonly dubbed the cage model. Initially proposed by Arhel and co-workers [72], this model proposes that given the limited number of RT molecules per virion, estimated to be between 50 and 150 [210], the crowded environment inside the capsid [211], and the multiple strand transfer reactions required for reverse transcription, an enclosed space, such as the viral core, would effectively concentrate all the components, enhancing the efficiency of the reaction.

The ability of reverse transcription to proceed in assembled capsids is also supported by a study on the role of the kinesin 1 heavy chain KIF5B protein in capsid transport through the microtubule network. Microtubules enhance uncoating, and deletion of this protein delays it, but without affecting synthesis of reverse transcription products, at least at early time points after infection [212].

DNA click-labelling with EdU combined with immunofluorescence microscopy has also detected the strong association of CA with newly synthesised reverse transcription products, again supporting the ability of viral DNA to be synthesised in capsid or capsid-like structures [121,122]. Another immunofluorescence study combined viral particle tracking with fractionation of viral capsids based on their size (the fate of capsid assay) and found that intact or nearly intact capsids support reverse transcription [107]. The Pathak lab has developed a method to directly monitor viral uncoating by constructing a GFP-CA chimeric protein that is incorporated into viral capsids in low quantities. This allowed the researchers to detect assembled viral capsids where reverse transcription is undergoing [103,104].

The reverse scenario, wherein reverse transcription is required for uncoating, has also been examined. In recent years, mounting evidence has supported this notion [Figure 4]. An early hint of this phenomenon came from a study combining the fate of capsid assay with ERT reactions carried out during increased periods of time, showing that the synthesis of the last proviral DNA fragment [the DNA flap] was necessary for uncoating [72]. Following this study, immunofluorescence tracking of viral particles combined with time-of-escape inhibition from TRIMCyp also led to the conclusion that reverse transcription contributes to uncoating [73], a notion supported by further work from the same group, which even established that the inhibition of first-strand transfer delays uncoating [93]. In addition, the fate of capsids where reverse transcription is inhibited showed that these capsids were more stable than those undergoing viral DNA synthesis [74].

The Rousso lab has studied the stability of isolated capsids undergoing reverse transcription through atomic force microscopy (AFM). Initially, they found that an increase of core stiffness occurs 7 h after the initiation of reverse transcription, followed by an abrupt fall. Interestingly, they detected an opening at the narrow end of viral capsids 12–17 h into reverse transcription, probably coinciding with the reaction end, and followed shortly after by further CA loss [213]. The addition of the capsid-stabilizing molecule PF74 led to the earlier appearance of the stiffness peak and opening at the capsid’s narrow end, which was not followed by further CA loss [214]. In ERT reactions carried out in the presence of IP6, they detected stiffness spikes concomitantly with specific stages of reverse transcription, followed by accelerated disassembly of capsids [205,207]. Specifically, reactions performed by an RNase H mutant RT unable to extend the DNA after the formation of minus-strand strong-stop showed only the first spike, while spikes were not recorded in reactions carried out in the presence of RT inhibitor efavirenz, which is consistent with the inability of the capsid to open in these conditions [205]. 

A study employing different CA mutants with increased or decreased stability, and an array of biochemical and cellular methods to determine capsid integrity, identified that a step after the first-strand transfer initiates capsid uncoating [215]. Analysis of the loss of association between a tetrameric CypA-DsRed reporter and the viral core has been used as a surrogate for capsid disassembly, and this reaction is accelerated by the initiation of reverse transcription [216]. Recently, in experiments tracking dually labelled particles capable of detecting the loss of capsid integrity by direct tagging of CA through amber codon suppression, it was observed that the addition of the RT inhibitor nevirapine produced a delay in uncoating [109].

Christense et al. employed transmission electron microscopy (TEM) to visualize the assembly state of capsids undergoing ERT for extended periods of time (up to completion of reverse transcription) and identified many capsids that had lost local patches of CA instead of suffering a complete disassembly of the lattice [Figure 4]. Interestingly, they observed loops of DNA protruding out of these capsid holes and proposed that the internal pressure exerted by the growing DNA produces these cracks [197]. In a recent study, the Pathak group, employing again their chimeric GFP-CA constructs, has observed that uncoating of the viral capsid requires the synthesis of dsDNA of over 3.5 Kb, linking again the size of the transcript with uncoating activity [198].

The concept that reverse transcription can proceed (and maybe finish) inside closed cores, together with the possibility of intact or nearly intact capsids reaching the nucleus, opened the possibility that reverse transcription could be completed in the cell nucleus. This is in contrast to early reports detecting full-length viral DNA in the cytoplasm [72,217,218]. By expressing the nuclear pore channel protein NUP62 fused to a drug-inducible dimerization domain, Dharan and co-workers were able to block nuclear pores and therefore the import of viral replication complexes. By adding RT inhibitors, the authors were able to show that the virus remains sensitive to them hours after becoming insensitive to nuclear pore blockade, indicating that reverse transcription continues after nuclear entry [102].

In addition, a large number of microscopy-based studies using different combinations of viral particle tagging, DNA labelling, and biochemical approaches have concluded that the synthesis of viral DNA finalizes inside the nuclear compartment. Therefore, by combining immunofluorescence microscopy and DNA click-labelling, a study found viral RNA in the nucleus of infected cells capable of serving as a template for reverse transcription [219]. Employing their CypA-DsRed system, and the time-of-escape from RT inhibitors, the Melikyan lab has detected ongoing reverse transcription in the nucleus [101]. Recently, the same lab used viral particles containing CA subunits labelled by amber codon suppression, and bearing a fluid marker (YFP-IN), to show that reverse transcription precedes uncoating and that both processes are completed inside the nucleus [109]. Similarly, other studies tracking cores labelled by incorporated GFP-CA subunits coupled with analysis of time-of-escape from nevirapine have concluded that reverse transcription is completed in the nucleus [103,104].

The ANCHOR system, enabling the direct labelling of viral DNA products, combined with fluorescence microscopy has also been employed for the detection of active reverse transcription complexes in the nucleus [105,115]. By identifying CA in nuclear fractions of infected cells earlier than the peak of reverse transcription, Guedán and co-workers determined that reverse transcription is completed in the nucleus [117]. In a similar approach, the fate of the capsid assay was used to detect high-order CA assemblies from nuclear fractions of infected cells, together with most reverse transcription intermediates, which the authors interpreted as an indication of nuclear reverse transcription [107].

Finally, some studies have established a connection between reverse transcription and the nuclear import of viral capsids. Early works found that reverse transcription was necessary for the nuclear import of viral replication complexes, based on the required presence of the DNA flap [72,88,218]. More recently, Dharan et al. observed that nevirapine treatment delayed the accumulation of CA in the nucleus and proposed that reverse transcription promotes nuclear import [102]. On the contrary, most recent studies could not detect changes in the efficiency or kinetics in the nuclear import of replication complexes, and therefore determined these two processes are independently regulated [96,98,99,101,119,120,123,220].

## 7. Conclusions

Reverse transcription and capsid uncoating are two intricate processes. Technical limitations on their study in living cells have hindered our understanding and led to erroneous conclusions over the years. The advent of more sensitive techniques, together with the ability to single out infecting viral particles, and key discoveries such as the role of IP6 or the ability of nuclear pores to dilate, has greatly propelled the development of the field. In addition, it has prompted authors to revisit old ideas, which more frequently than not has resulted in a deep modification of existing models.

Nevertheless, there are many questions that remain to be answered. Some of them are (i) where does capsid uncoating start? (ii) where does it finish? (iii) how is the capsid shed during uncoating? (iv) what state of reverse transcription is required, if any, to trigger uncoating? (v) is capsid opening, even if partially, needed for completion of reverse transcription? (vi) where is reverse transcription completed? (vii) is there a single mechanism for capsid uncoating, or are alternative routes possible (as it seems to occur with N74D or P90A CA mutant viruses [102])?

Only by deeply understanding the viral and cellular processes underpinning HIV-1 replication can new therapeutic avenues be discovered, as is exemplified by the recent FDA approval of the capsid inhibitor lenacapavir.

## Figures and Tables

**Figure 1 ijms-25-07167-f001:**
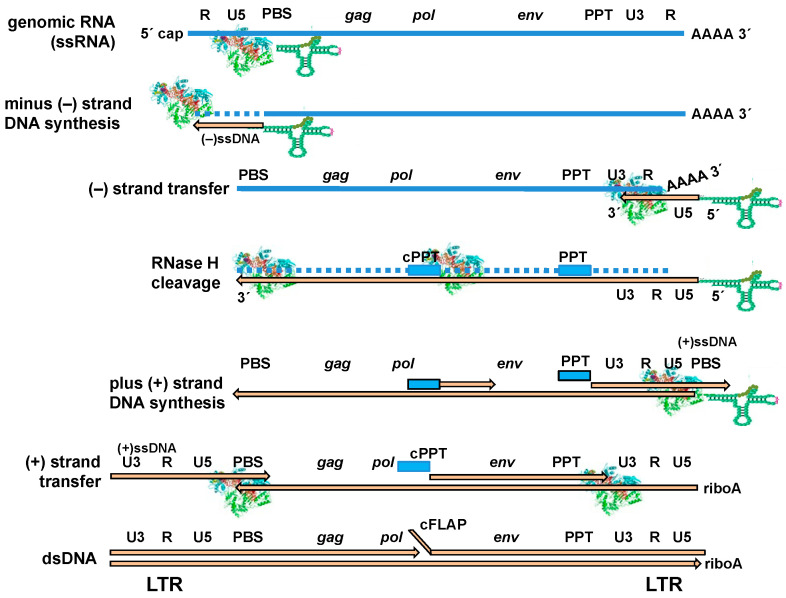
Outline of the HIV-1 reverse transcription process. The steps involving the conversion of single-stranded genomic RNA into dsDNA are shown. RNA is shown in blue and DNA in light orange. A ribbon diagram is used to represent the HIV RT structure. Abbreviations used: LTR, long terminal repeat; PBS, primer binding site; PPT, polypurine tract; R, repeat; U3, unique 3′; U5, unique 5′. Adapted from Menéndez-Arias et al. [31], with permission from Elsevier.

**Figure 2 ijms-25-07167-f002:**
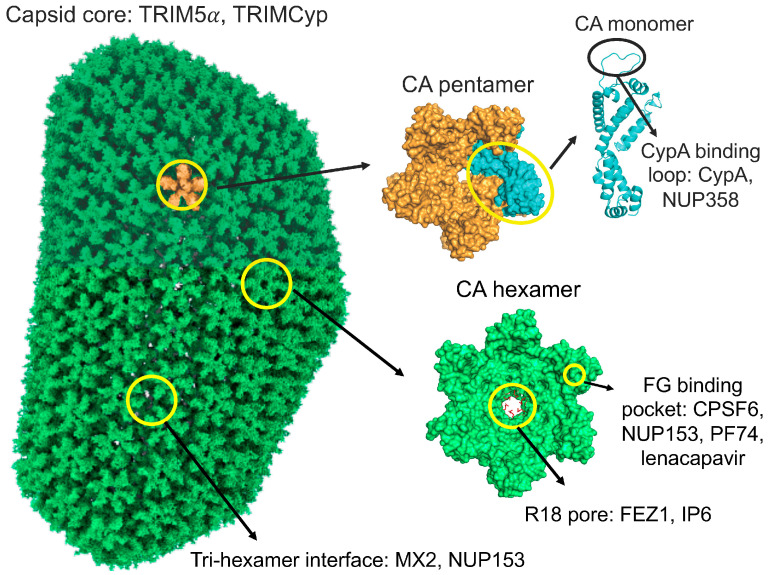
Structure of HIV-1 capsid, interacting interfaces, and main binders. Complete model of the mature HIV-1 capsid shell, encompassing 186 hexamers and 12 pentamers based on PDB file 3J3Y [86], and designed using the ChimeraX free software [87]. An individual CA pentamer is highlighted in orange and depicted on the right (PDB file 3P05 [88]), together with an individual hexamer (PDB file 4U0D, [89]) in green. A monomeric CA (PDB file 4XFY, [90]) has been highlighted in cyan in the pentamer and is depicted next to it. The main interacting interfaces on each capsid structure have been indicated with a circle, together with the main cellular proteins and small molecules interacting with them: (i) CypA binding loop: CypA and NUP358; (ii) FG pocket: NUP153, CPSF6, PF74, and lenacapavir; (iii) tri-hexamer interface: MX2 and NUP153; (iv) R18 pore: FEZ1 and IP6; (v) capsid core: TRIM5α and TRIMCyp.

**Figure 3 ijms-25-07167-f003:**
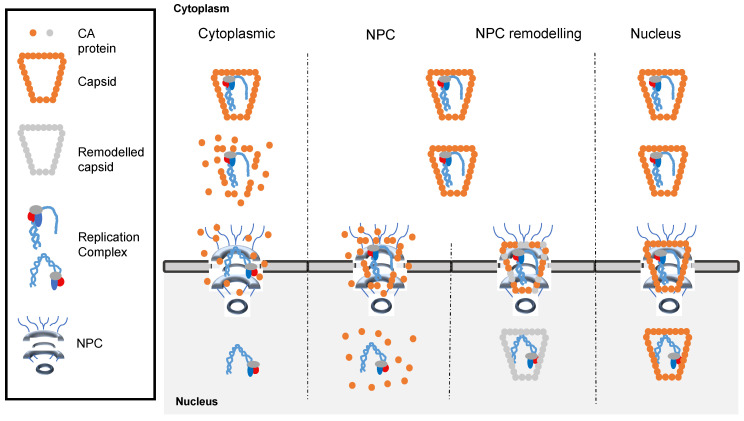
Models for HIV-1 capsid uncoating. From left to right: uncoating early on the cytoplasm; uncoating at the nuclear pore; capsid remodelling during nuclear translocation followed by nuclear uncoating; uncoating inside the nuclear lumen. All four models are depicted, with the unmodified capsid, remodelled capsid, nuclear pore complex (NPC), and two different replication complexes illustrating the different stages of viral DNA synthesis shown.

**Figure 4 ijms-25-07167-f004:**
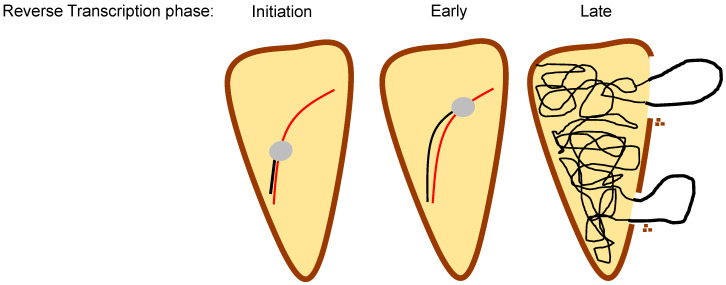
Effect of reverse transcription on capsid uncoating. An HIV-1 core containing the reverse transcription complex is represented, with the RNA in red, the growing DNA in black, and the RT as a grey oval. Different stages of reverse transcription are depicted, from initiation, where only a few nucleotides have been incorporated into the DNA strand, to late stages, where the viral DNA has been extended. According to most recent studies [197,205,213,214], progression of reverse transcription to its late stages triggers the loss of capsid integrity, creating holes in the capsid lattice from where DNA loops protrude, as depicted in the right-most figure.

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
