# Peer review of "May I Help You with Your Coat? HIV-1 Capsid Uncoating and Reverse Transcription"

_ijms, 2024, doi:10.3390/ijms25137167_

Round 1

Reviewer 1 Report

Comments and Suggestions for Authors

This is a very interesting and thorough review on HIV capsid uncoating and retrotranscription by Arribas et al. The manuscript has a catchy title, the text is very complete, detailed and balanced, giving the reader a broad overview on our current understanding of the HIV capsid uncoating and retrotranscription. As noted in this review, both processes are intertwined and have been extensively studied by many different groups and many studies have yielded controversial (or different) results as noted in this review.  The authors explained the different uncoating models and also gave a nice overview of cellular factors that affect RT. The topic is of interest, the text is well referenced, and I liked that the work has several figures.

I only have some minor comments for the authors:

Line 62: it reads as if all ART drugs target the RT; I suggest the authors rephrase this sentence for enhanced accuracy.

Line 187: Authors mentioned the rhesus macaque TRIM5a but I suggest the authors rephrase that sentence since there’s also TRIM5 in humans.

Figure 2 could benefit from a legend that explains what is shown, and also includes the definition of PPT.

Figure 4 could also benefit from a legend or some text with arrows to better illustrate the contents of the figure.

Please confirm the acknowledgments section is correct.

Reviewer 2 Report

Comments and Suggestions for Authors

The article « May I help you with your coat? HIV-1 capsid uncoating and reverse transcription” by Arribas and coworkers reviews the recent advances in the understanding of the timing of HIV-1 capsid disassembly and a possible role of reverse transcription in this phenomenon.

This review is very interesting, well written, and it summarizes several recent developments in our understanding of HIV-1 uncoating.

Therefore, I only have a couple of minor suggestions that could improve this manuscript:

1-      The fact that NUPs are nucleoporins is never clearly stated. However, it is an important point as NUPs interact with CA hexamers and at CA hexamer interfaces: this supports the fact that assembled capsids are directly interacting with nuclear pores.

2-      Similarly, the fact that PF74 and lenacapavir are CA inhibitors stabilizing the capsid should be stated from the start (line 220), as it supports further the importance of the uncoating in the replication cycle.

3-      §5 is interesting, but is mostly an inventory of cellular partners involved in the regulation of RT. However, it gives no information on if, and how, these factors could interfere with the association between RT and uncoating, which is the topic of this review. Therefore, this paragraph should be removed.

Reviewer 3 Report

Comments and Suggestions for Authors

Journal: International Journal of Molecular Sciences 

Manuscript ID: ijms-3071071 

Type of manuscript: Review 

Title: May I help you with your coat? HIV-1 capsid uncoating and reverse 

transcription  

Authors: Laura Arribas, Luis Menéndez-Arias, Gilberto Betancor *

Molecular Microbiology

This work focuses on the timing of reverse transcription and the state of the viral genome at the time of entry into the cell nucleus, among the extensive and complex series of viral replication processes.

The authors summarize the information obtained from the development of measurement equipment and experimental methods over the past 15 years. The conclusion is that the model shown by the authors as Figure 3 is not the leftmost model. Althogh the experimental method is carefully described, but this reviewer feels that the description of the detected object is insufficient in the references. As a review, it is somewhat unfriendly, so please rearrange it partially with respect to the following points. This reviewer would like to recommend that this paper be published as a review in International Journal of Molecular Sciences after making sufficient necessary corrections.

1. Please add an explanation of each component to the legend of Figure 1. Specifically,

(a) spell out "PPT"

(b) For L98-L163, please specify which row in Figure 1 the description is referring to

(c) Make it clear that the ribbon model is RT.

2. Required spelling out or explanation

L174 "CA" capsid

L218 "FG" Or is it the same as "phenylalanyl-glycin" in L295?

L218 "NTD", "CTD" N-terminal, C-terminal domain

3. About Figure 2

The contents of i)-v) (L216-L225) in the text and i)-v) (L233-L235) in the legend are indicated in a mixed way, which is very confusing for readers. Please unify it into one or the other.

4. About Figure 3

There are two types of cartoon for "Replication Complex". Please add a sentence somewhere to explain the difference. For L259-L340, please separate the subtitles in a way that corresponds to the classification in Figure 3. This reviewer speculates that up to L319, "NPC" and "NPC remodelling" are mixed, and from L320 onwards, "Nucleus" corresponds.
